# Common Genetic Variations Involved in the Inter-Individual Variability of Circulating Cholesterol Concentrations in Response to Diets: A Narrative Review of Recent Evidence

**DOI:** 10.3390/nu13020695

**Published:** 2021-02-22

**Authors:** Mohammad M. H. Abdullah, Itzel Vazquez-Vidal, David J. Baer, James D. House, Peter J. H. Jones, Charles Desmarchelier

**Affiliations:** 1Department of Food Science and Nutrition, Kuwait University, Kuwait City 10002, Kuwait; mohamad.abdullah@ku.edu.kw; 2Richardson Centre for Functional Foods & Nutraceuticals, University of Manitoba, Winnipeg, MB R3T 6C5, Canada; itzel_vazquez25@yahoo.com; 3United States Department of Agriculture, Agricultural Research Service, Beltsville, MD 20705, USA; david.baer@usda.gov; 4Department of Food and Human Nutritional Sciences, University of Manitoba, Winnipeg, MB R3T 2N2, Canada; j_house@umanitoba.ca; 5Nutritional Fundamentals for Health, Vaudreuil-Dorion, QC J7V 5V5, Canada; peterjones@nfh.ca; 6Aix Marseille University, INRAE, INSERM, C2VN, 13005 Marseille, France

**Keywords:** lipids, gene-diet interaction, personalized nutrition, single nucleotide polymorphism, genetic variant, cardiovascular diseases, nutrigenetics

## Abstract

The number of nutrigenetic studies dedicated to the identification of single nucleotide polymorphisms (SNPs) modulating blood lipid profiles in response to dietary interventions has increased considerably over the last decade. However, the robustness of the evidence-based science supporting the area remains to be evaluated. The objective of this review was to present recent findings concerning the effects of interactions between SNPs in genes involved in cholesterol metabolism and transport, and dietary intakes or interventions on circulating cholesterol concentrations, which are causally involved in cardiovascular diseases and established biomarkers of cardiovascular health. We identified recent studies (2014–2020) that reported significant SNP–diet interactions in 14 cholesterol-related genes (*NPC1L1, ABCA1, ABCG5, ABCG8, APOA1, APOA2, APOA5, APOB, APOE, CETP, CYP7A1, DHCR7, LPL,* and *LIPC*), and which replicated associations observed in previous studies. Some studies have also shown that combinations of SNPs could explain a higher proportion of variability in response to dietary interventions. Although some findings still need replication, including in larger and more diverse study populations, there is good evidence that some SNPs are consistently associated with differing circulating cholesterol concentrations in response to dietary interventions. These results could help clinicians provide patients with more personalized dietary recommendations, in order to lower their risk for cardiovascular disease.

## 1. Introduction

Personalized nutrition is the next frontier for the food and health industries. Presently, consumers report high expectations for the role proper genetic information could play in their dietary choices for the prevention and treatment of chronic diseases [1,2]. Cardiovascular disease (CVD) is still the leading cause of death globally, and it is estimated that 90% of cases are preventable, with healthy diet considered as the first line of intervention [3,4]. Lowering of blood lipid concentrations is a major target in both the primary and secondary prevention of CVD, but high interindividual variability exists in response to any given dietary intervention [5].

Genome-wide association studies (GWAS) have identified numerous single nucleotide polymorphisms (SNPs) that are associated with the variability of fasting blood cholesterol concentrations. These genetic variations have been estimated to explain about 30% of total variance [6,7]. Yet, the application of the results to personalized dietary recommendations is not straightforward because these studies usually do not consider the effect of diet. Several clinical trials have investigated the relationship between dietary interventions and blood lipid concentrations, considering the genetic characteristics of the participants. However, by design, they usually only focus on individual SNPs, resulting in a relatively low explained genetic variance [8,9]. A greater part of the variability in blood lipid concentrations could be explained by the additive effects of several SNPs, which, taken individually, may only have small, and barely significant, effects [10,11]. While numerous studies have reported SNPs that are significantly associated with blood cholesterol concentrations in response to various diets [8], there is a need to review additional data generated in this field. 

The objective of this narrative review is, thus, to summarize recent findings of interactions between individual SNPs in major cholesterol-related genes and dietary intakes, relative to shaping circulating cholesterol concentrations. The review also highlights studies that used combinations of SNPs to increase the explained variability in cholesterol concentrations following dietary interventions. Looking at the sum of evidence accumulated in the field enables discussion of what is still missing in order to put this knowledge into practice.

## 2. Search Process and Criteria

For the purpose of this review, we considered only genes that are explicitly involved in cholesterol absorption, metabolism, and transport pathways (Figure 1). A systematic search was utilized, and gene–diet interaction studies relative to the genes of interest were retrieved, with no restriction on study design, dietary protocol, population, or quality throughout the preliminary search. Searches were performed on articles published between 1 January 2014 (as a follow-up to our previous review on the topic [8]) and 30 September 2020 using the MEDLINE (PubMed) database, and searching string variations of the following keywords: (diet OR gene-diet OR dietary) AND (cholesterol) AND (SNP OR polymorphism OR genetic variant). A total of 291 articles were identified and examined individually to exclude research articles that did not report statistically significant cholesterol responses to gene–diet interactions, did not present clear associations, or did not provide data details. With the exclusion of all review articles, animal studies, and articles in languages other than English, as well as those considering factors other than diet, such as physical activity, alcohol intake, smoking, and others, a total of 21 distinct research articles were eligible for inclusion in the individual SNP–diet interaction analysis (Section 3), and four articles were eligible for the combinatory patterns of SNPs analysis (Section 4) below.

The following sections present statistically significant findings regarding the effects of interactions between SNPs (taken individually or in combinatory patterns) and dietary intakes on circulating cholesterol concentrations in observational studies and dietary interventions. Nomenclature for all genes in this work is reported according to international standards of the Human Genome Organization (HUGO) [12], with description of variants at the DNA level and reference to SNPs as rs#-xx homozygotes or rs#-x alleles, even when authors of some of the reviewed articles reported variants as defined by their encoding amino acid changes, using the 3-letter code or the 1-letter code, with no mention of the rs# for their SNPs. Where applicable in the sections below, the meanings of such amino acid codes are addressed.

## 3. Studies Investigating the Effect of SNPs Taken Individually

In this section, a summary of the recent evidence on statistically significant gene–diet interactions is presented considering individual SNP data (Table 1). This approach has been the most widely utilized in research and reported in the literature due to both technical and statistical reasons. For instance, sequencing and genotyping techniques, such as commercially available SNP assays and PCR methods, are presently low-cost and easy-to-implement. Moreover, dietary interventions are typically expensive to carry out, and usually have relatively low sample sizes, resulting in limited statistical power. Nonetheless, they allow for identification of new genetic associations based on known or predicted SNP functions. Importantly, the SNP–diet interactions identified in clinical settings also constitute good candidates for building genetic risk scores (GRS) of cumulative SNPs, aiming at predicting practical changes in biomarkers of health in response to dietary intakes.

### 3.1. SNPs in Genes Encoding Transporters Involved in Cholesterol Absorption

NPC1 like intracellular cholesterol transporter 1 (NPC1L1) is an apical multi-pass membrane protein that mediates the transport of sterols into enterocytes and hepatocytes through clathrin-mediated endocytosis, and is essential for cholesterol trafficking [34,35,36]. It is now well-established that dietary plant sterols (PS) can inhibit intestinal cholesterol absorption through several mechanisms, such as competition for incorporation into mixed micelles, hydrolysis of esters by digestive enzymes, enterocyte uptake by NPC1L1, and incorporation into chylomicrons [34,37]. In a randomized crossover supplementation study involving 19 post-menopausal females, participants carrying the CC genotype at rs2072183 (a synonymous variant with leucine at 272, published as L272L) in *NPC1L1* showed significantly different changes in serum TC and LDL-C concentrations (0.38 ± 0.29 and 0.33 ± 0.29 mmol/L, respectively) compared to carriers of the G allele (−0.29 ± 0.10, and −0.18 ± 0.12 mmol/L, respectively) after consuming for four weeks a beverage providing 1.5 g PS/d [16]. Of note, these differences were no longer significant when an outlier in the participants homozygous for the minor allele was excluded from the statistical analysis.

ATP-binding cassette subfamily A1 (ABCA1) is a basolateral membrane protein that transports cholesterol and phospholipids out of enterocytes and hepatocytes, which are then collected by apolipoprotein A1 (encoded by *APOA1*) to form nascent HDL particles [38,39]. A 2015 cross-sectional study involving 1598 premenopausal Mexican females found that serum HDL-C concentrations were negatively correlated with carbohydrate intake (*r* = −0.362), and positively correlated with fat intake (*r* = 0.357), in carriers of the T allele at rs9282541 in *ABCA1* (published as RC/CC in the R230C missense variant, as defined by the amino acid change) but not in those carrying the CC genotype (published as RR in R230C variant) [13]. This variant is restricted to the Americas, where its presence explains up to 4% of the variation in HDL-C concentrations [40]. In an intervention study, 56 healthy Chinese adults received a washout diet where carbohydrate contributed 54% of energy for 7 d, followed by a high-carbohydrate/low-fat diet where carbohydrate and fat contributed 70% and 14% of energy, respectively, for 6 d [14]. Males carrying the A (published as K) allele and females carrying the GG (published as RR) genotype at rs2230806 in *ABCA1* (G1051A; published as R219K, a missense variant, as defined by the amino acid change) displayed lower LDL-C/HDL-C concentration ratios (1.08 ± 0.27 vs. 1.48 ± 0.64 in males, and 1.01 ± 0.18 vs. 1.37 ± 0.33 in females) after the high-carbohydrate/low-fat diet [14].

The *ATP*-binding cassette subfamily G member 5 and 8 (*ABCG5/G8*) genes encode two transporters (ABCG5 and ABCG8) that are responsible for the excretion of sterols by hepatocytes and enterocytes [41]. Rare and common genetic variations in *ABCG5/G8* have been linked to increased intestinal absorption of cholesterol and PS [42,43,44]. In a 2016 randomized crossover study by our group, involving 101 normocholesterolemic adults, the *ABCG5* rs6720173-GG homozygotes had higher serum TC (change: 0.18 ± 0.06 vs. −0.07 ± 0.07 mmol/L) and LDL-C (change: 0.17 ± 0.05 vs. −0.06 ± 0.07 mmol/L) concentrations relative to rs6720173-C allele carriers after three servings/d of dairy (low-fat milk, low-fat yogurt, and regular cheddar cheese) versus a dairy-free diets for 4 wk each [15]. These data replicate, to some extent, previous associations described among Spanish children [45], where male carriers of the *ABCG5* rs6720173-G allele had higher plasma TC and LDL-C concentrations with a daily intake of 14.3–34.1 g saturated fatty acids (SFA) compared to CC homozygotes [45]. Another crossover multiple-dose supplementation study with beta-cryptoxanthin and PS in healthy postmenopausal females showed that carriers of the CC genotype of *ABCG8* rs6544718 (published as A632V, a missense variant, as defined by the amino acid change) had lower serum TC concentrations (−0.34 ± 0.09 mmol/L) compared to carriers of the T allele (0.08 ± 0.20 mmol/L) after consuming a beverage providing 1.5 g/d of PS plus 750 µg beta-cryptoxanthin over 4 wk [16].

### 3.2. SNPs in Genes Encoding Apolipoproteins

*Apolipoprotein A1* (*APOA1*) encodes a protein that drives maturation of HDL particles [46,47]. A weight loss intervention of one arm involving 82 obese patients showed that carriers of the GG genotype at rs670 in *APOA1* had lower serum HDL-C concentrations compared to A allele carriers, both at baseline (1.43 ± 0.25 vs. 1.50 ± 0.24 mmol/L) and after 12 wk (1.37 ± 0.21 vs. 1.45 ± 0.23 mmol/L) on a hypocaloric diet (reduction of 500 kcal/d) [17]. A similar 12 wk hypocaloric intervention involving 282 obese participants, by the same group, reported improvements in HDL-C concentrations among the rs670-A allele carriers versus GG homozygotes at baseline, as well as after both high-fat (1.42 ± 0.23 vs. 1.27 ± 0.18 mmol/L) and low-fat (1.53 ± 0.21 vs. 1.40 ± 0.23 mmol/L) diets [18].

*Apolipoprotein A2 (APOA2)* encodes the second most common protein found in HDL particles [48]. A 2016 cross-sectional study involving 697 diabetic participants from Tehran, Iran, reported that carriers of the *APOA2* rs5082-CC genotype who, based on a semi-quantitative food frequency questionnaire (FFQ) analysis, had higher SFA intake (>28.5 g/d, considering mean SFA intake of the study population as cutoff point) and displayed a higher LDL-C/HDL-C concentration ratio (2.27 ± 0.72 vs. 2.04 ± 0.63) compared to carriers of the T allele [19]. 

*Apolipoprotein A5* (*APOA5*) encodes an essential protein component of several lipoproteins such as VLDL, HDL, and chylomicrons [49]. A number of meta-analyses have demonstrated that SNPs in *APOA5* may be associated with increased risk for CVD, where, in particular, carriers of the *APOA5* rs662799-C allele exhibit higher concentrations of TC and TG, and lower concentrations of HDL-C compared to non-carriers [50,51]. In a cross-sectional study conducted among young Mexicans (18–25 y), participants carrying the rs662799-C allele and who traditionally consumed a diet higher in PUFAs, with median (percentile 25–75) of 10 (8–12) g/d vs. 8 (6–12) g/d also showed lower serum HDL-C concentrations compared to TT homozygotes (1.1 (0.9–1.2) mmol/L vs. 1.2 (1.0–1.4) mmol/L); although this was not reported by the authors as part of their gene–diet interaction data [20]. In another cross-sectional study involving 1128 premenopausal Korean females, among participants who had total energy intakes ≥2001 kcal/d, based on a 24-h recall method and a semi-quantitative FFQ data analysis, rs662799-CC homozygotes displayed lower serum HDL-C concentrations (~1.16 vs. 1.42 mmol/L) compared to T allele carriers [21].

*Apolipoprotein B* (*APOB*) encodes a protein that modulates the formation and blood clearance of LDL particles and triglyceride (TG)-rich lipoproteins. Circulating apoB-48 and apoB-100 concentrations are well-known biomarkers of cardiovascular health [52]. A cross-sectional study including 6470 Korean adults showed that among those who reported consuming higher levels of energy (≥1983.7 kcal/d), based on a semi-quantitative FFQ analysis, carriers of the minor allele (G) at rs1469513 in *APOB* had higher plasma TC (+2.42%) and LDL-C (+1.96%) concentrations compared to the AA homozygotes; similar TC response was seen with higher fat intake (≥19.9% energy), where the difference between G allele carriers and AA homozygotes was 2.04% [22]. In contrast, homozygotes for the A allele who consumed carbohydrates in excess of 64.9% of energy exhibited higher TC (+1.81%) and LDL-C (+2.80%) concentrations compared to G allele carriers [22].

Apolipoprotein E is one of the major proteins involved in the transport of cholesterol and TG within the circulation [53]. A 2017 retrospective analysis of 120 participants enrolled in the DIVAS study, a parallel design based on a food-exchange model of SFA, monounsaturated fatty acid (MUFA), and polyunsaturated fatty acid (PUFA) intakes in free-living individuals for 16 wk, reported that only *APOE* rs1064725-TT homozygotes, but not G allele carriers, showed a reduction in serum TC concentrations after the MUFA diet (−0.71 ± 1.88 mmol/L) compared to the SFA (0.34 ± 0.55 mmol/L) or n-6 PUFA diets (−0.08 ± 0.73 mmol/L) [23]. A German diabetes study evaluated the interaction between dietary patterns, using FFQ data, and *APOE* polymorphisms in 348 diabetic patients, where among carriers of the APOE ε2 isoform (E2/E2, E2/E3; rs429258: TT, rs7412: TT/CT), lower versus higher consumption frequencies of butter, cream cake, French fries, or alcoholic beverages were independently associated with a 40% reduction in serum LDL-C concentrations [24]. In a randomized crossover design involving 63 mildly hypercholesterolemic individuals who consumed 2 g/d of PS for 28 d, carriers of the APOE ε4 isoform had a greater reduction in serum LDL-C concentrations (−0.31 ± 0.07 mmol/L vs. −0.13 ± 0.05 mmol/L) compared to carriers of the APOE ε3 isoform [25]. This study also reported an interaction between *APOE* and *CYP7A1*-rs3808607, where participants of all genosets, except for *CYP7A1*-rs3808607 T/T + APOE ε3, showed reductions in LDL-C concentrations in response to PS consumption [25]. Furthermore, in a secondary analysis of data from the RISCK study, a 24-wk five-arm parallel intervention involving 389 adults, carriers of E4 relative to E3/E3 showed greater decreases in plasma TC concentrations (−0.28 ± 0.13 mmol/L) when SFA was replaced with low glycemic index (GI) carbohydrate on a lower fat diet, and a relative increase in TC concentrations (0.30 ± 0.14 mmol/L) when SFA was replaced with MUFA and high GI carbohydrates [26].

### 3.3. Genes Encoding Additional Proteins Involved in the Cholesterol Metabolic Pathway

Cholesteryl ester transfer protein (CETP) exchanges TG from lipoproteins (VLDL or LDL) with cholesteryl esters from HDL particles [54]. In the CORDIOPREV clinical trial involving 424 Spanish participants with metabolic syndrome (MetS) who received a Mediterranean diet rich in fat from olive oil (35% fat, 22% MUFA) or a low-fat diet (28% fat, 12% MUFA), carriers of the minor allele (T) at rs3764261 in *CETP* showed higher plasma HDL-C concentrations (1.06 ± 0.03 vs. 0.98 ± 0.02 mmol/L) compared to GG homozygotes after 1 y on the Mediterranean diet, but not after the low-fat diet [27]. The *CETP* rs3764261–diet interaction was more recently investigated among 4700 adults from the Tehran Lipid and Glucose Study, a population-based prospective design, where among carriers of the T (published as A) allele, serum TC concentrations decreased from 8.02 to 5.58 mmol/L as fish intake increased from <3.6 g/d (quartile 1) through >14.9 g/d (quartile 4) compared to GG (published as CC) homozygotes, after 3.6 years of follow-up [28].

Cholesterol 7-alpha hydroxylase, encoded by *CYP7A1*, is a rate-limiting enzyme in the classic bile acid synthesis pathway in the liver. *CYP7A1* rs3808607 has been associated with inter-individual variability in circulating cholesterol concentrations in response to dietary intakes. For instance, after 4 wk on a crossover design with dairy versus dairy-free diets, we have previously shown that carriers of the rs3808607-G allele had higher serum TC concentrations (ranging between 0.20 ± 0.06 and 0.28 ± 0.12 mmol/L for GT and GG genotypes, respectively) compared to TT homozygotes (−0.11 ± 0.08 mmol/L) [15]. In response to PS intake, however, the rs3808607-GG homozygotes showed greater a reduction in serum LDL-C concentrations (−0.46 ± 0.12 vs. −0.05 ± 0.07 mmol/L) compared to the TT homozygotes [25]. In yet another crossover design involving 30 mildly hypercholesterolemic adults who received a breakfast that contained either 3 g of high molecular weight barley β-glucan or a control diet daily for 5 wk each, carriers of the rs3808607-G allele showed greater reduction in serum TC concentrations (−0.58 and −0.70 mmol/L for GT and GG genotypes, respectively) compared to TT homozygotes (−0.31 mmol/L) [29].

The 7-dehydrocholesterol reductase, encoded by *DHCR7*, is the enzyme that catalyzes the terminal reaction of cholesterol synthesis in the liver. In response to dairy intervention, carriers of the *DHCR7* rs760241 minor allele (A) showed higher serum LDL-C concentrations (0.26 ± 0.08 vs. 0.06 ± 0.05 mmol/L) relative to GG homozygotes [15]. To our knowledge, this study by members of our group was the first, and thus far only, evidence of such interaction between the *DHCR7* SNP rs760241 and dietary intake.

Lipoprotein lipase, encoded by *LPL*, is an enzyme catalyzing the hydrolysis of TG from apoB-containing lipoproteins. A 2014 dietary intervention involving 56 healthy young Chinese showed that only male carriers of the *LPL* rs326 minor allele (G) had an increase in plasma HDL-C concentrations (1.32 ± 0.27 vs. 1.21 ± 0.21 mmol/L) after versus before a 6-d diet of high-carbohydrate (70.1% energy)/low-fat (13.7% energy) [30]. In the cross-sectional Chennai Urban Rural Epidemiological Study, involving type 2 diabetes cases and controls from India, carriers of the *LPL* rs1121923 minor allele (T) exhibited higher serum HDL-C concentrations (1.2 vs. 1.1 mmol/L) compared to CC homozygotes after consumption of a higher fat diet (3rd tertile: 28.4 ± 2.5% energy), as assessed by a semi-quantitative FFQ [31]. No differences were observed among genotypic groups consuming a diet providing less than 23.5% of energy from fat.

Hepatic lipase, encoded by *LIPC*, is an enzyme involved in reverse cholesterol transport. In a 2017 crossover, randomized controlled trial involving 42 Caribbean Hispanics, carriers of the *LIPC* rs1800588 major allele (C) had higher plasma HDL-C concentrations (1.3 ± 0.03 vs. 1.1 ± 0.04 mmol/L during phase 1, and 1.4 ± 0.03 vs. 1.2 ± 0.03 mmol/L during phase 2 of the trial) after consuming a high-fat Western diet (39% energy) compared to a low-fat traditional Hispanic diet (20% energy) for 4 wk each, whereas no difference was observed between Western and Hispanic diets among TT homozygotes [32]. In the POUNDS LOST trial, a long-term weight-loss intervention with diets varying in macronutrient composition, overweight and obese carriers of the *LIPC* rs2070895 minor allele (A) showed an increase in serum TC concentrations (β ± SE: 0.19 ± 0.07 mmol/L) on a high-fat diet (40% energy) and a slight decrease in HDL-C concentrations (β ± SE: −0.04 ± 0.02 mmol/L) on a low-fat diet (20% energy) compared to rs2070895-GG homozygotes at 2 yr of the intervention [33]. 

## 4. Combinatory Patterns of SNPs Influencing Changes in Blood Cholesterol Concentrations in Response to Dietary Interventions

Recently, the availability of low-cost genetic testing kits and new bioinformatic tools has allowed significant progress in understanding the complex interplay between genetics and diet. Development of GRS, in which various SNPs information is aggregated, offers a valuable approach to explain a higher percentage of variability in blood cholesterol profiles compared to analyzing individual SNPs [10,11]. 

In a study published in 2015, Justesen et al. investigated the effect of the interaction between GRS and lifestyle factors, including diet, on fasting serum lipid concentrations in a population-based Danish cohort comprised of 5961 male and female participants (mean age = 46.1 y, mean BMI = 26.3 kg/m^2^) [55]. GRS of increased TC (74 loci), LDL-C (58 loci), or triglyceride (TG) (39 loci), and decreased HDL-C (71 loci) concentrations were built based on SNPs previously identified following GWAS of circulating fasting lipid concentrations. Dietary habits were estimated following dietary questionnaires and were classified as unhealthy, moderately healthy, and healthy. There was no significant effect of GRS and dietary habit interaction on baseline fasting serum cholesterol concentrations. The effect of specific components of the diet was not investigated.

In a secondary analysis of data from the Food4Me study, a multi-country European online randomized controlled intervention exploring the potential for personalized nutrition to enhance nutritional and health outcomes, the authors built a GRS based on 14 SNPs previously identified in GWAS of components of MetS and that showed a significant association with any component of MetS at baseline of the intervention [56]. Thus, most selected SNPs had no existing association with cholesterol metabolism and transport. They found that after 6 months of intervention, among the 1263 male and female participants (mean age = 40.8 y, mean BMI = 25.4 kg/m^2^), those with a low GRS exhibited a significantly higher decrease in TC concentrations, compared to those with a high GRS (approximately −0.1 mmol/L), independently of Mediterranean diet adherence level.

In a secondary analysis of data from a randomized crossover study [15], members of our group demonstrated a combinatory action of three SNPs in cholesterol-related genes, where among 101 normocholesterolemic participants (age = 18–69 y), carriers of the genotypic combination of *ABCG5* rs6720173-C, *CYP7A1* rs3808607-TT, and *DHCR7* rs760241-GG exhibited a maximal reduction in serum LDL-C concentrations relative to a combination of rs6720173-GG, rs3808607-G, and rs760241-A genotypes (change: −0.37 ± 0.12 vs. 0.38 ± 0.14 mmol/L), following a blended dairy (3 servings/d for 4 wk) versus dairy-free intervention [57].

In a 2019 study, Guevara-Cruz et al. investigated the effect of the interaction between a GRS and a dietary intervention on serum HDL-C concentrations in 67 Mexican adults (20–60 y) with a BMI ≥ 25 kg/m^2^ and who had at least three out of five positive criteria for MetS [58]. The dietary intervention consisted in the consumption for 2.5 months of a diet following the National Cholesterol Education Program-Adult Treatment Panel III (NCEP-ATP III) guidelines, i.e., low in saturated fats (<7% energy) and cholesterol (<200 mg/d), with 20–30 g/d fiber and a 25% reduction in total energy intake. The GRS consisted of six SNPs previously associated with MetS in GWAS, lifestyle intervention studies, and meta-analyses, and that were associated with changes in serum HDL-C concentrations following dietary intervention. The GRS built explained 42% of the variance in HDL-C concentrations following the NCEP-ATP III diet, and participants with a low GRS displayed an increase (0.08 mmol/L) in HDL-C concentrations, whereas participants with a high GRS had a decrease in HDL-C concentrations (−0.08 mmol/L). Additionally, these results were validated in an independent clinical cohort of 89 participants with MetS following a similar dietary intervention.

Collectively, these studies show that the additive effect of several SNPs, which taken individually may have small independent effects, could explain a greater part of the variability in response to a dietary intervention when combined. These data show the advantage of complex analyses including combinatory patterns of SNPs compared to the relatively low predictive value of individual SNP approaches. Although these models are promising for personalized dietary advice, the results are variable and require validation in controlled dietary interventions, including different age groups, gender, and across ethnicities.

## 5. Conclusions

This review highlights the most recent findings from studies investigating the effects of SNP–diet interactions on blood cholesterol concentrations, where 20 SNPs in 14 cholesterol-related genes showed significant associations with various dietary intakes. This review also underscores important progress in the field, given the emerging scientific literature on the effect of combinations of SNPs on blood cholesterol concentrations in interaction with dietary intakes. This approach is likely to increase the explained variability in such a complex phenotype, and offers a valuable insight into the relationship between diet and CVD risk.

Dietary interventions designed to lower circulating cholesterol concentrations, such as those limiting intakes of trans fat from processed foods, or replacing saturated fatty acids with unsaturated fatty acids, for example, are established means for reducing the risk of developing CVD, regardless of a person’s genetic makeup. Carriers of certain genotypes within cholesterol-related genes may, however, respond far better than others to a given dietary intervention, possibly leading to an improved cardiovascular health outcome. For instance, a reduction of approximately 10% in LDL-C concentrations was observed in adults carrying the genotypic combination of *ABCG5* rs6720173-C, *CYP7A1* rs3808607-TT, and *DHCR7* rs760241-GG, relative to the combination of rs6720173-GG, rs3808607-G, and rs760241-A genotypes, after dairy versus dairy-free diets [57]. With evidence suggesting at least 1% reduction in CVD incidence or mortality with each 1% reduction in LDL-C or TC concentrations [12,59,60], the clinical impact of such nutrigenetic studies is obvious.

Still, in addition to factors such as dietary components and population groups, it is imperative that the type of genetic mutation, whether missense, intron, or other, is also considered when assessing the potential impacts on the gene products. Of the individual SNPs reported in this review, six variants are listed by the Single Nucleotide Polymorphism Database (dbSNP) as missense variants (*ABCA1* rs9282541 and rs2230806, *ABCG5* rs6720173, *ABCG8* rs6544718, *APOE* rs7412, and *DHCR7* rs760241), thus describing changes in single nucleotides that result in different amino acid sequences, while others include intron, upstream transcript, synonymous, 5 prime UTR, or 3 prime UTR variants. Considering the type of genetic mutation may provide some mechanistic insight into the interpretation and discussion of findings in association studies, especially when detailed reports on the molecular mechanisms of action in the field are rarely available.

Future research, including multi-site intervention trials, should take such clinical and mechanistic aspects into consideration and, furthermore, explore combinatory patterns of genetic variability in order to better understand the impact of gene–diet interactions on serum lipid profiles, and the potential cardiovascular impacts thereof. The individual SNPs reported in this review, and those previously reported [8], are good starting candidates to build GRS aiming at predicting changes in blood cholesterol concentrations in response to dietary interventions. As an additional tool to advance the field, the availability of artificial intelligence and machine-learning approaches for identifying panels of SNPs, which in combination could better predict an individual’s lipid response to dietary interventions, offer tremendous potential [61]. The latter methods may allow a more rapid advancement of the field of personalized nutrition.

## Figures and Tables

**Figure 1 nutrients-13-00695-f001:**
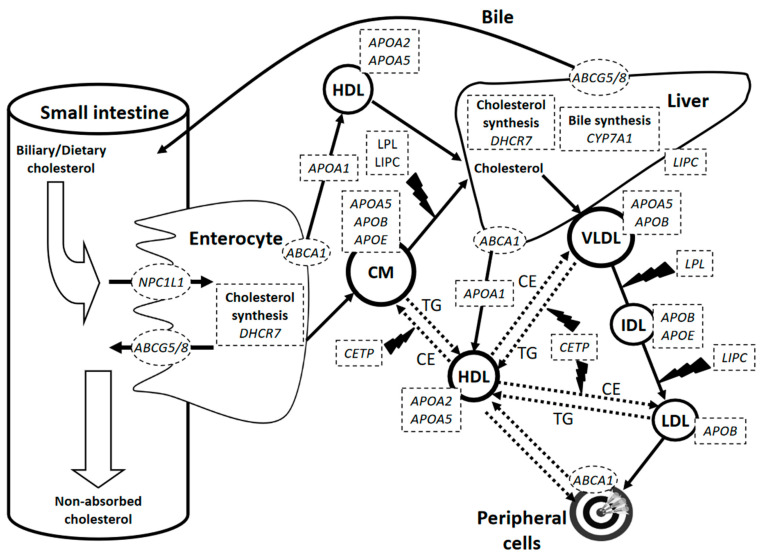
Summary of genes of lipid metabolism and transport pathways involved in the variability of circulating cholesterol concentrations in response to diets. Genes displayed are those for which significant SNP–diet interactions have been reported recently (2014–2020). *ABCA1*, *ATP binding cassette subfamily A member 1*; *ABCG5/8*, *ATP binding cassette subfamily G member 5/8*; APOA1, *apolipoprotein A1*; *APOA2*, *apolipoprotein A2*; *APOA5*, *apolipoprotein A5*; *APOB*, *apolipoprotein B*; *APOE*, *apolipoprotein E*; CE, cholesteryl esters; *CETP*, *cholesteryl ester transfer protein*; *CYP7A1*, *cholesterol 7-alpha-hydroxylase*; *DHCR7*, *7-dehydrocholesterol reductase*; HDL, high-density lipoprotein; IDL, intermediate-density lipoprotein; LDL, low-density lipoprotein; *LPL*, *lipoprotein lipase*; *LIPC*, *hepatic lipase*; *NPC1L1*, *NPC1 like intracellular cholesterol transporter 1*; TG, triglycerides; VLDL, very-low-density lipoprotein

**Table 1 nutrients-13-00695-t001:** Summary of epidemiological and dietary intervention studies reporting significant effects of individual SNP–diet interactions on circulating cholesterol concentrations.

Author	Gene	SNP	Study Design	Diet	Population and Ethnicity	Major Cholesterol Outcomes
Jacobo-Albavera et al. (2015) [13]	*ABCA1*	rs9282541	Cross-sectional	Dietary CHO and fat intakes	1598 premenopausal females, Mexicans	HDL-C concentrations negatively correlated with CHO intake and positively correlated with fat intake in T allele carriers but not CC homozygotes
Liu et al. (2014) [14]	*ABCA1*	rs2230806	Dietary intervention	7-d of washout diet followed by 6-d of high-CHO/low-fat diet	56 healthy adults, Chinese	Lower LDL-C/HDL-C concentration ratios in A allele male carriers and GG female homozygotes after vs. before high-CHO/low-fat diet
Abdullah et al. (2016) [15]	*ABCG5*	rs6720173	Crossover	Dairy vs. dairy-free diets	101 healthy adults, Canadians	Higher TC and LDL-C concentrations in GG homozygotes vs. C allele carriers after 3 servings/d of dairy vs. dairy-free diets
Granado-Lorencio et al. (2014) [16]	*ABCG8*	rs6544718	Crossover	750 μg β-cryptoxanthin and 1.5 g/d PS, single and combined	19 postmenopausal females, Spanish	Lower TC concentrations in CC homozygotes vs. T allele carriers after β-cryptoxanthin + PS combined intake
de Luis et al. (2018) [17]	*APOA1*	rs670	Dietary intervention	Hypocaloric diet of one arm	82 obese adults, Spanish	Lower HDL-C concentrations in GG homozygotes vs. A allele carriers at baseline and after hypocaloric diet
de Luis et al. (2019) [18]	*APOA1*	rs670	Dietary intervention	Hypocaloric high-fat vs. low-fat diets	282 obese adults, Spanish	Higher HDL-C concentrations in A allele carriers vs. GG homozygotes at baseline and after both high-fat and low-fat diets
Noorshahi et al. (2016) [19]	*APOA2*	rs5082	Cross-sectional	Dietary SFA intake (>28.5 g/d), assessed by FFQ	697 type 2 diabetic adults, Iranians	Higher LDL-C/HDL-C concentration ratio in CC homozygotes vs. T allele carriers with higher SFA intake
Dominguez-Reyes et al. (2015) [20]	*APOA5*	rs662799	Cross-sectional	Dietary fat intake, assessed by FFQ	200 young normal-weight and obese adults, Mexicans	Lower HDL-C concentrations and higher PUFA intake in C allele carriers vs. TT homozygotes
Lim et al. (2014) [21]	*APOA5*	rs662799	Cross-sectional	Dietary intake, assessed by 24-h recall and FFQ	1128 premenopausal females, Koreans	Lower HDL-C concentrations in CC homozygotes with higher total energy intake (≥2001 kcal/d)
Doo et al. (2015) [22]	*APOB*	rs1469513	Cross-sectional (‘KoGES’ Study)	Total energy and macronutrient intake, assessed by FFQ	6470 adults, Koreans	Higher TC and LDL-C concentrations in G allele carriers with higher energy or fat intake, and in AA homozygotes with higher CHO intake
Shatwan et al. (2017) [23]	*APOE*	rs1064725	16-wk parallel dietary intervention (‘DIVAS’ Study)	Isoenergetic diets rich in SFA, MUFA, or n-6 PUFA	120 adults with moderate cardiovascular risk, Caucasians	Lower TC concentrations only in TT homozygotes after MUFA-rich diet vs. SFA-rich or n-6 PUFA-rich diets
Weber et al. (2016) [24]	*APOE*	rs429258 rs7412	Cross-sectional (German Diabetes Study)	Dietary fat intake, assessed by FFQ	348 diabetics, Germans	Lower LDL-C concentrations in *APOE* ε2 carriers with lower vs. higher intake frequencies of butter, cream cake, French fries, or alcoholic beverages
Mackay et al. (2015) [25]	*APOE*	rs429258 rs7412	Crossover	2 g/d of PS	63 mildly hypercholesterolemic adults, Canadians	Greater reduction in LDL-C concentrations after PS intake in *APOE* ε4 vs. *APOE* ε3 carriers
Griffin et al. (2018) [26]	*APOE*	rs429258 rs7412	24-wk five-arm parallel dietary intervention (‘RISCK’ Study)	Replacing SFA with either MUFA or carbohydrate of high or low GI	389 adults at increased risk of developing MetS, white Caucasians	Greater decreases in TC concentrations in carriers of E4 vs. E3/E3 when SFA was replaced with low GI carbohydrate on a lower fat diet, and an increase in TC concentrations when SFA was replaced with MUFA and high GI carbohydrates
Garcia-Rios et al. (2016) [27]	*CETP*	rs3764261	1-y dietary intervention (‘CORDIOPREV’ Study)	Mediterranean diet (35% fat, 22% MUFA) vs. low-fat diet (28% fat, 12% MUFA)	424 MetS subjects, Spanish	Higher HDL-C concentrations after Mediterranean diet in T allele carriers vs. GG homozygotes
Hosseini-Esfahani et al (2019) [28]	*CETP*	rs3764261	Population-based prospective design (Tehran Lipid and Glucose Study)	Usual dietary intake, assessed by FFQ	4700 adults, Iranians	Lower TC concentrations and higher fish intakes in T allele carriers vs. GG homozygotes after 3.6 years of follow-up
Abdullah et al. (2016) [15]	*CYP7A1*	rs3808607	Crossover	Dairy vs. dairy-free diets	101 healthy adults, Canadians	Higher TC concentrations in G allele carriers vs. TT homozygotes after 3 servings/d of dairy vs. dairy-free diets
Mackay et al. (2015) [25]	*CYP7A1*	rs3808607	Crossover	2 g/d of PS	63 mildly hypercholesterolemic adults, Canadians	Greater reduction in LDL-C concentrations after PS intake in GG vs. TT homozygotes
Wang et al. (2016) [29]	*CYP7A1*	rs3808607	Crossover	Barley β-glucan vs. control diet	30 mildly hypercholesterolemic adults, Canadians	Lower TC concentrations after 3 g/d of high molecular weight barley β-glucan in G allele carriers vs. TT homozygotes
Abdullah et al. (2016) [15]	*DHCR7*	rs760241	Crossover	Dairy vs. dairy-free diets	101 healthy adults, Canadians	Higher LDL-C concentrations in A allele carriers vs. GG homozygotes after 3 servings/d of dairy vs. dairy-free diets
Zhu et al. (2014) [30]	*LPL*	rs326	Dietary intervention	7-d of washout diet followed by 6-d of high-CHO (70% energy)/low-fat (~14% energy) diet	56 healthy Chinese Han youth	Higher HDL-C concentrations in G allele male carriers after vs. before high-CHO/low-fat diet
Ayyappa et al. (2017) [31]	*LPL*	rs1121923	Cross-sectional (Chennai Urban Rural Epidemiological Study)	Dietary intakes, assessed by FFQ	788 type 2 diabetes cases and 1057 controls, Asian Indians	Higher HDL-C concentrations in T allele carriers with high fat diet vs. CC homozygotes
Smith et al. (2017) [32]	*LIPC*	rs1800588	Crossover	Comparing a high-fat Western diet and a low-fat traditional Hispanic diet	42 adults, Caribbean Hispanic descent	Higher HDL-C concentrations in CC homozygotes after high-fat Western diet (39% energy) vs. low-fat traditional Hispanic diet (20% energy)
Xu et al. (2015) [33]	*LIPC*	rs2070895	2-y randomized weight-loss dietary intervention (‘POUNDS LOST’ Study)	Dietary intakes	743 overweight or obese adults, multiethnic groups	Higher TC concentrations after high fat intake, and higher HDL-C concentrations after low fat intake, in A allele carriers vs. GG homozygotes
Granado-Lorencio et al. (2014) [16]	*NPC1L1*	rs2072183	Crossover	750 μg β-cryptoxanthin and 1.5 g/d PS, single and combined	19 postmenopausal females, Spanish	Higher TC concentrations in CC homozygotes vs. G allele carriers after PS intake only

**Abbreviations**: ABC, ATP-binding cassette subfamily; APO, apolipoproteins; *CETP*, cholesteryl ester transfer protein; CHO, carbohydrate; *DHCR7*, 7-dehydrocholesterol reductase; *CYP7A1*, cholesterol 7-alpha-hydroxylase; FFQ, food frequency questionnaire; GI, glycemic index; HDL-C, high-density lipoprotein cholesterol; LDL-C, low-density lipoprotein cholesterol; *LPL*, lipoprotein lipase; *LIPC*, hepatic lipase; MetS, metabolic syndrome; MUFA, monounsaturated fatty acids; SFA, saturated fatty acids; SNP, single nucleotide polymorphism; PUFA, polyunsaturated fatty acids; *NPC1L1*, NPC1 like intracellular cholesterol transporter 1; PS, plant sterols, TC, total cholesterol.

## Data Availability

Not applicable.

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
