# Peer review of "Common Genetic Variations Involved in the Inter-Individual Variability of Circulating Cholesterol Concentrations in Response to Diets: A Narrative Review of Recent Evidence"

_nutrients, 2021, doi:10.3390/nu13020695_

Round 1
Reviewer 1 Report
The manuscript titled; Common Genetic Variations Involved in the Inter-individual 2 Variability of Circulating Cholesterol Concentrations in Response to Diets: A Narrative Review of Evidence is
Serving its written aim to " present recent findings concerning the effects of the interactions between SNPs in genes involved in cholesterol metabolism and transport and dietary intakes or interventions on circulating cholesterol concentrations, which are causally involved in cardiovascular diseases or established biomarkers of cardiovascular health." This review is a continuous review for this group of authors other similar excellent reviews in this subject (Nutrition Reviews73(8):523–543, 2015, Current Cardiology Reports (2019) 21:38). As this is a hot topic with constant update a narrative review is suitable, however the authors should emphasis the specific progress done since the latest reviews (theirs or others), with additional detailed focus on new concepts such as the combined SNP score.
- The word recent should be added to the title to read" …..A narrative review of recent evidence"
- Abstract: specify the time frame (2015-2020)
- As the authors one of the conclusions is that an additive and combined SNPs could explain a higher proportion of variability in response to dietary interventions, they should add their recommendation/point of view for the specific SNPs that their additive response should be taken into consideration. This information could also be presented in a table.
- Of all the studies using combined GRS ,how much (percent) the combined score explain the variance in cholesterol level. Please add to the relevant studies.
Author Response
Reviewer 1 :
Comments and Suggestions for Authors:
The manuscript titled; Common Genetic Variations Involved in the Inter-individual Variability of Circulating Cholesterol Concentrations in Response to Diets: A Narrative Review of Evidence is serving its written aim to " present recent findings concerning the effects of the interactions between SNPs in genes involved in cholesterol metabolism and transport and dietary intakes or interventions on circulating cholesterol concentrations, which are causally involved in cardiovascular diseases or established biomarkers of cardiovascular health." This review is a continuous review for this group of authors other similar excellent reviews in this subject (Nutrition Reviews 73(8):523–543, 2015, Current Cardiology Reports (2019) 21:38). As this is a hot topic with constant update a narrative review is suitable, however the authors should emphasis the specific progress done since the latest reviews (theirs or others), with additional detailed focus on new concepts such as the combined SNP score.
We thank the Reviewer for having reviewed our manuscript and we would like to answer the points raised as follows.
- The word recent should be added to the title to read “[…] A narrative review of recent evidence”.
This has been modified accordingly.
- Abstract: specify the time frame (2015-2020)
This has been modified accordingly.
- As the authors one of the conclusions is that an additive and combined SNPs could explain a higher proportion of variability in response to dietary interventions, they should add their recommendation/point of view for the specific SNPs that their additive response should be taken into consideration. This information could also be presented in a table.
Which SNPs will increase the explained variance in blood lipid profiles when combined, together with other SNPs or host-related factors, cannot be predicted but needs to be measured. The different SNPs we report in this review and those we have reported in previous reviews are good candidates but some others are also likely to contribute. We have modified the Conclusion to add our recommendation for future studies as follows:
“Individual SNPs reported in this review or previously reported [8] constitute good starting candidates to build GRS aiming at predicting changes in blood cholesterol concentrations in response to dietary interventions.”
- Of all the studies using combined GRS, how much (percent) the combined score explain the variance in cholesterol level. Please add to the relevant studies.
Of the three studies reporting a significant GRS diet interaction, only one (Guevara-Cruz et al.) calculated the variance explained by the GRS: 42% of the variance of the change in serum HDL cholesterol concentration following the NCEP-ATP III diet was explained by the GRS. This piece of information was already included in the first version of the manuscript. For the other studies, it is unfortunately impossible for us to calculate the explained variance based on the published data.
Reviewer 2 Report
This review briefly summarizes studies on the relationships between the presence of SNPs in certain genes related with cholesterol metabolism, diet and the interindividual variability towards cardiovascular diseases. In particular, it empashizes the potential interaction between different SNPs to explain this variability.
The manuscript would be improved if the authors address the following issues:
- In section 2, SNPs found in certain genes are compiled (also in Table 1). In some cases the consequence in the corresponding gene product (e.g. a given missense mutation) is indicated. I would appreciate if the authors describe more extensively these potential effects in the gene products: whether a missense mutation is known (i.e. experimentally investigated) or predicted to be deleterious, or whether a given SNPs is expected to affect splicing, etc....This would provide more mechanistic insight into the presentation and discussion of association studies.
- Also in section 2, the different genes are recopilated and the function of the corresponding protein is indicated. These are clustered in three groups (note that section 2.2 is found twice; clearly this is a typo). It would be good for the reader to integrate the functions of different gene product in a figure summarizing the roles of these in cholesterol metabolism.
- In some cases such as ABCA1, there seems to be some ambiguity regarding the consequence of the SNPs (e.g. "G1051A; published as R219K"). Could the authors, when necessary, to address the meaning of these inconsistencies?
Author Response
Reviewer 2 :
Comments and Suggestions for Authors:
This review briefly summarizes studies on the relationships between the presence of SNPs in certain genes related with cholesterol metabolism, diet and the interindividual variability towards cardiovascular diseases. In particular, it empashizes the potential interaction between different SNPs to explain this variability.
We thank the Reviewer for having reviewed our manuscript and we would like to answer the points raised as follows.
The manuscript would be improved if the authors address the following issues:
- In section 2, SNPs found in certain genes are compiled (also in Table 1). In some cases the consequence in the corresponding gene product (e.g. a given missense mutation) is indicated. I would appreciate if the authors describe more extensively these potential effects in the gene products: whether a missense mutation is known (i.e. experimentally investigated) or predicted to be deleterious, or whether a given SNPs is expected to affect splicing, etc....This would provide more mechanistic insight into the presentation and discussion of association studies.
We agree with the Reviewer on the importance of such description of the potential effects in the gene products. However, given the scarcity of detailed studies on the molecular mechanisms of action reported in the field, reaching specific conclusions based on outcomes of the wide range of gene x diet interaction studies currently available is a complicated process, even when the type of mutation is known. Still, while considering this, a general statement on the clinical implications of findings and potential mechanistic insights is now given in the Conclusions (lines 313-336), as follows:
“Dietary interventions designed to lower the circulating cholesterol concentrations, such as those limiting intakes of trans fat from processed foods, or replacing saturated fatty acids with unsaturated fatty acids, for example, are an established means for reducing the risk of developing CVD, regardless of a person’s genetic makeup. Carriers of certain genotypes within cholesterol-related genes may, however, respond far better than others to a given dietary intervention, possibly leading to an improved cardiovascular health outcome. For instance, approximately 10% reduction in LDL-C concentrations was observed in adults carrying the genotypic combination ABCG5 rs6720173-C, CYP7A1 rs3808607-TT, and DHCR7 rs760241-GG, relative to the combination rs6720173-GG, rs3808607-G, and rs760241-A, after dairy versus dairy-free diets [55]. With evidence suggesting at least 1% reduction in CVD incidence or mortality with each 1% reduction in LDL-C or TC concentrations, the clinical impact of such nutrigenetic studies is likely meaningful.
Still, in addition to factors such as the dietary components and population groups, it is imperative that the type of genetic mutation, whether missense, intronic, or others, be also considered when assessing the potential impacts on the gene products. Of the individual SNPs reported in this review, 6 variants are listed by the Single Nucleotide Polymorphism Database (dbSNP) as missense variants (ABCA1 rs9282541 and rs2230806, ABCG5 rs6720173, ABCG8 rs6544718, APOE rs7412, and DHCR7 rs760241), thus describing changes in a single nucleotide that result in a different amino acid sequence, while others as intron, upstream transcript, synonymous, 5 prime UTR, or 3 prime UTR variants. Considering this may provide some mechanistic insight into the interpretation and discussion of findings in association studies, especially when detailed reports on the molecular mechanisms of action in the field are rarely available.”
- Also in section 2, the different genes are recopilated and the function of the corresponding protein is indicated. These are clustered in three groups (note that section 2.2 is found twice; clearly this is a typo). It would be good for the reader to integrate the functions of different gene product in a figure summarizing the roles of these in cholesterol metabolism.
The typo has been corrected. We have taken the Reviewer’s remark into account and we have added a figure illustrating the role of the proteins for which SNPs in their encoding genes are reported in this review in circulating cholesterol metabolism and transport.
- In some cases such as ABCA1, there seems to be some ambiguity regarding the consequence of the SNPs (e.g. "G1051A; published as R219K"). Could the authors, when necessary, to address the meaning of these inconsistencies?
Nomenclature for all genes has been reported according to international standards (HUGO), with description of variants at the most basic DNA level and reference to SNPs as rs#-xx homozygotes or rs#-x alleles. Some authors continue to choose to report variants as defined by their encoding amino acid changes, using the 3-letter code (e.g. Arg, Lys) or the 1-letter code (e.g. R, K). We now highlight this in the Introduction (lines 66-72) and, where applicable throughout the text, also briefly define the meaning of such amino acid changes as reported by authors of the original articles, as follows:
“Nomenclature for all genes in this work is reported according to international standard (HUGO), with description of variants at the DNA level and reference to SNPs as rs#-xx homozygotes or rs#-x alleles, even when authors of some of the reviewed articles reported variants as defined by their encoding amino acid changes, using the 3-letter code or the 1-letter code, with no mention of the rs# for their SNPs. Where applicable in the sections below, the meanings of such amino acid codes are addressed.”
Round 2
Reviewer 2 Report
The manuscript has been improved properly.
